# The Importance of Time and Place: Nutrient Composition and Utilization of Seasonal Pollens by European Honey Bees (*Apis mellifera* L.)

**DOI:** 10.3390/insects12030235

**Published:** 2021-03-10

**Authors:** Gloria DeGrandi-Hoffman, Vanessa Corby-Harris, Mark Carroll, Amy L. Toth, Stephanie Gage, Emily Watkins deJong, Henry Graham, Mona Chambers, Charlotte Meador, Bethany Obernesser

**Affiliations:** 1Carl Hayden Bee Research Center, USDA Agricultural Research Service, 2000 East Allen Road, Tucson, AZ 85719, USA; vanessa.corby@usda.gov (V.C.-H.); Mark.Carroll@USDA.GOV (Mark Carroll); Emily.Watkinsdejong@USDA.GOV (E.W.d.); Henry.Graham@USDA.GOV (H.G.); Mona.Chambers@USDA.GOV (Mona Chambers); Charlotte.Meador@USDA.GOV (C.M.); 2Department of Entomology, Iowa State University, 2310 Pammel Drive, 339 Science Hall II, Ames, IA 50011, USA; amytoth@iastate.edu; 3Georgia Institute of Technology, School of Physics, Howey Physics Building, 837 State Street NW, Atlanta, GA 30313, USA; sgage7@gatech.edu; 4Department of Entomology, University of Arizona, Forbes 410, P.O. Box 210036, Tucson, AZ 85721, USA; bobernesser@email.arizona.edu

**Keywords:** *Apis**mellifera* nutrition, nutrients in pollen, fat body, hypopharyngeal glands, *hex 70*, *hex 110*, *vg* expression

## Abstract

**Simple Summary:**

Honey bees rely on pollen and nectar to provide nutrients to support their yearly colony cycle. Specifics of the cycle differ among geographic regions as do the species of flowering plants and the nutrients they provide. We examined responses of honey bees from two different queen lines fed pollens from locations that differed in floral species composition and yearly colony cycles. We detected differences between the queen lines in the amount of pollen they consumed and the size of their hypopharyngeal glands (HPG). There were also seasonal differences between the nutrient composition of pollens. Spring pollens collected from colonies in both locations had higher amino and fatty acid concentrations than fall pollens. There also were seasonal differences in responses to the pollens consumed by bees from both queen lines. Bees consumed more spring than fall pollen, but digested less of it so that bees consumed more protein from fall pollens. Though protein consumption was higher with fall pollen, HPG were larger in spring bees.

**Abstract:**

Honey bee colonies have a yearly cycle that is supported nutritionally by the seasonal progression of flowering plants. In the spring, colonies grow by rearing brood, but in the fall, brood rearing declines in preparation for overwintering. Depending on where colonies are located, the yearly cycle can differ especially in overwintering activities. In temperate climates of Europe and North America, colonies reduce or end brood rearing in the fall while in warmer climates bees can rear brood and forage throughout the year. To test the hypothesis that nutrients available in seasonal pollens and honey bee responses to them can differ we analyzed pollen in the spring and fall collected by colonies in environments where brood rearing either stops in the fall (Iowa) or continues through the winter (Arizona). We fed both types of pollen to worker offspring of queens that emerged and open mated in each type of environment. We measured physiological responses to test if they differed depending on the location and season when the pollen was collected and the queen line of the workers that consumed it. Specifically, we measured pollen and protein consumption, gene expression levels (*hex 70*, *hex 110*, and *vg*) and hypopharyngeal gland (HPG) development. We found differences in macronutrient content and amino and fatty acids between spring and fall pollens from the same location and differences in nutrient content between locations during the same season. We also detected queen type and seasonal effects in HPG size and differences in gene expression between bees consuming spring vs. fall pollen with larger HPG and higher gene expression levels in those consuming spring pollen. The effects might have emerged from the seasonal differences in nutritional content of the pollens and genetic factors associated with the queen lines we used.

## 1. Introduction

Nutritional landscapes change as plants flower and seed in seasonal succession, providing the range of nutrients required to support the activities of herbivores throughout the year. Nutrients required during periods of reproduction and growth differ from those for overwintering especially if resources are sparse. Dietary needs of animals and their availability in the environment link organisms to the ecosystems where they reside, and affect their life span, reproduction [1], migration [2,3], and spread as invasive species [4].

An organism with a seasonal cycle that can provide a model for connections between changing nutritional needs and nutrient availability is the honey bee. Honey bees obtain their nutrients from pollen and nectar [5]. In undisturbed ecosystems, the nutrients are provided by a seasonal progression of flowering plants. Pollen supplies proteins, lipids, vitamins and minerals [5], and nectar provides primarily carbohydrates [6]. Honey bees of all ages feed on nectar, but pollen is consumed primarily by workers during the first 7 days after adult eclosion, when they serve as nurse bees that feed and care for larvae [7]. Nurse bees convert the pollen they consume to nutrient-rich jelly in specialized organs called hypopharyngeal glands (HPG) [7,8]. Young larvae and the queen consume jelly.

As colonies go through their yearly cycle, they implement seasonal strategies to support growth, reproduction, and preparation for overwintering. In the spring and much of the summer, nurse bees consume pollen, grow large HPG and vigorously rear brood so colonies can expand and reproduce through swarming. If colonies are in northern latitudes or temperate regions where winter temperatures are low, brood rearing ceases in late fall and does not resume until late January or early February. Bees remain confined in the hive, but remain active by forming a winter cluster [9]. Bees thermoregulate the winter cluster by metabolically converting honey stores to heat [10,11,12]. Bees that comprise the winter cluster are reared from late August to October [13,14] and differ physiologically from spring and summer bees. Winter bees have low juvenile hormone levels [15], high vitellogenin titers [15,16], and hypertrophied HPG [15,17] that have less developed organelles, reduced protein expression patterns and lower metabolic activity than HPG in spring bees [18]. In southern latitudes however, where winter temperatures are warmer, colonies continue to forage and can rear some brood throughout the winter.

The annual cycles of colonies require nutritional support from seasonal pollens [18,19]. There is some evidence of a connection between nutrients in pollens and seasonal activities of honey bees. For example, spring pollens can have higher concentrations of protein and amino and fatty acids that support brood rearing and colony expansion (e.g., tryptophan, valine, serine, and omega-3 for HPG development and worker jelly production), while fall pollens can be rich in nutrients needed for overwintering and nest thermoregulation (e.g., proline) [19]. The nutrient profiles of spring and summer pollen mixtures over regions where colonies are established might be similar because colony expansion through brood rearing in the spring is a general strategy for colony survival. Nutrients in fall pollens however, might be more variable because overwintering strategies differ depending on colony location. For example, fall pollens in temperate regions might have high levels of lipids that can be stored in fat bodies of bees during winter confinement.

How colonies respond to the nutritional profile of landscapes where they establish also could be affected by genetic factors based on queen lines or location where queens are reared and mated. Local adaptations to biotic and abiotic conditions can occur within regional populations and improve long-term survival and establishment [20]. There is evidence from Europe that honey bee colonies with locally adapted stock have greater overwintering survival than non-local stock [21,22]. A study in the U.S. however, indicated that queen stock or region of origin did not affect colony growth and survival but rather the abundance and diversity of floral resources [23]. Differences in results between studies indicate the difficulties in discerning colony level effects of queen origin due to variability in environmental factors such as nutritional resources. There is evidence however, for fine-tuning of basic biochemical mechanisms in individual honey bees that may optimize metabolic capacities across the climatic regions where their colonies establish. For example, bees from colder climates where heat production and thermoregulation of the winter cluster are essential to survival have higher levels of proteins associated with the major energy generating pathways of the mitochondria than bees from warmer climates [24].

If metabolism is a target for local adaptations that enhance fitness, then nutrients available in seasonal pollens and the bees’ responses to them might differ based on the location where the pollen is collected and queens are reared and mated. To test this, we collected pollen in spring and fall from colonies located in areas where colony overwintering strategies differ (Iowa and Arizona). The pollen was fed to worker offspring from queens reared and open mated in areas where bees are either confined overwinter (Iowa) or can rear brood throughout the year (California). Nutrient composition of the pollens was evaluated and compared between seasons and locations. We fed the spring and fall pollens to bees that emerged in the same season as when the pollen was collected, and measured the bees’ responses. Measurements included consumption, protein digestion, hemolymph protein levels and size of the HPG. As expression levels of vitellogenin can differ with season especially in regions where there is winter confinement [13], we measured *vg* expression in the fat body. Expression of hexamerin 70a (*hex 70a)* and hexamarin 110 (*hex 110*) in the fat body also were measured because these genes play a role in nutrient storage, physiology, and overwintering or insect diapause [25,26].

Our measurements of nutrients in seasonal pollens and honey bee responses to them were used to address two questions. The first is whether nutrients available in pollens differ based on geographic location and season. Secondly, do workers that are offspring of queens reared and open-mated in different geographic locations consume and respond to pollen differently depending on season and region where the queens were sourced? If workers respond differently to pollen based on queen source, it may be due to genetic differences between the bees. However, if bees respond similarly to the same pollen, but differently based on the season and location where the pollen was collected, it would suggest environmental effects tied to nutrient content. Bees also could respond differently to pollens due to a combination of genetic and environmental effects. Information from this study can provide evidence for physiological responses tied to nutrient composition of pollens that may differ based on the species mixture collected by the bees and season when they were collected. Measuring the responses in bees sourced from regions where yearly colony cycles differ especially in overwintering could indicate whether metabolic responses are due to nutrients or are tempered by the environments where the bees were sourced.

## 2. Materials and Methods

### 2.1. General Overview

The study was conducted as a common garden experiment at the USDA-ARS Carl Hayden Bee Research Center in Tucson, Arizona, USA in 2016 with European honey bees. Queens reared and open-mated in either Iowa (IAq) or California (CAq) headed the experimental colonies. IAq were sourced from Fassbinder Apiaries, (Elgin, IA, USA), and were primarily of Carniolan stock (*Apis mellifera carnica*, Pollmann). CAq were from Pendell Apiaries (Stonyford, CA, USA) and were Cordovan Italian queens (*Apis mellifera ligustica*). We did not use queens from Arizona since the resident population is Africanized [27]. Fifteen queens from each source were marked with a paint dot on their thorax prior to introducing them to their colony. Two trials were conducted: one in the June (spring bees) and another in the October (fall bees).

Spring and fall Arizona pollen (AZp) was collected using pollen traps on six colonies spaced across apiaries located at the University of Arizona, West Campus Agricultural Center in Tucson, Pima County, Arizona, USA (AZp). Pollen from Iowa (IAp) was collected similarly from colonies at the Horticultural Research Station in Ames, Story County, Iowa, USA in the spring. The Iowa sites for fall pollen were the Horticultural Research Station and five prairies in the Chichaqua Bottoms Greenbelt, Polk County, Iowa. AZp was collected from February to April, 2015 (spring pollen) and from September to November, 2015 (fall pollen). IAp was collected in May and June, 2016 (spring pollen) and August and September, 2015 (fall pollen). Pollen was collected and immediately frozen at −20 °C until fed to bees.

Pollen for all trials was removed weekly from traps and pooled in 2-gallon plastic Ziplock bags. The pollen was mixed by shaking and rolling the bag. A random sample from each weekly bag was taken and pooled into a new 2-gallon plastic Ziplock bag. The pollen in the bag was mixed as described above. A one liter sample of the pollen was removed from the bag for feeding the bees in the cages (see below). The procedure was repeated for fall AZp and for pollen collected in Iowa.

### 2.2. Pollen Identification

ITS gene sequencing was used to identify spring and fall floral sources for IAp and AZp using methods described in [19]. Briefly, DNA from AZp or IAp was extracted from a 30 mg sample of ground and mixed spring and fall AZp and IAp used to feed bees in the cages (see below). The DNA was subjected to PCR using the ITS primers and PCR protocol described by [28]. The PCR product was cloned into a vector and at least 100 clones were directly sequenced. The cloned ITS gene was characterized to the family or genus level using NCBI’s BLAST algorithm [29].

### 2.3. Pollen Protein and Amino Acid Analyses

All analyses were conducted using the same ground pollen fed to bees in cages. Soluble protein in pollen was measured using three random samples of spring and fall pollens and analyzing them for soluble protein with a BCA protein assay (Thermo Scientific, Waltham, MA, USA) as described in [30]. The pollen was not dried prior to analysis to more closely capture protein concentrations in corbicular loads consumed by the bees.

Amino acids were quantified from six random samples of the spring or fall IAp and AZp using procedures described in [19]. We extracted and quantified amino acids from each pollen sample after fluoroalkyl chloroformate derivatization using an EZFAAST amino acid hydrolysate kit [31] (Phenomenex, Inc., Torrance, CA, USA). All essential amino acids except tryptophan and cysteine were characterized following acid hydrolysis. Glutamine and asparagine were converted by acidic conditions to their acid equivalents (glutamic and aspartic acid). As a result, each amine/acid pair was indistinguishable. Tryptophan was extracted after base hydrolysis, and characterized by fluoroalkyl chloroformate derivatization [32], and analyzed by EI GC–MS on an HP 7890A gas chromatograph coupled to a HP 5975D mass spectrometer detector (Agilent, Inc., Santa Clara, CA, USA). Cysteine residues were quantified after phenylisothiocyanate (PITC) derivatization adapted from [33]. Derivatized amino acids were characterized by comparison of mass spectra and retention times (RT) with derivatized authentic standards. The fraction of the total sample present in each injected sample was estimated from the amount of norvaline internal standard recovered.

### 2.4. Pollen Lipid and Fatty Acid Analyses

Total lipids in spring and fall IAp and AZp were quantified from four samples of each pollen type using a chromic acid oxidation assay as described in [19,34]. Fatty acids were analyzed based on six samples of each pollen type, using fatty acid methyl ester (FAME) analysis described in [19]. FAME compounds were elucidated by comparing mass spectra and retention times with esterified standards. FAME compounds were quantified by comparison of characteristic mass fragments (*m*/*z*) with known amounts of authentic standards. The fraction of the total sample injected was calculated from the amount of internal standard (pentadecanoic acid) detected in each injected sample.

### 2.5. Cage Setup, Pollen Feeding and Sampling

Our study began six weeks after queens were introduced into colonies to ensure sufficient numbers of offspring. Frames of sealed brood from all colonies headed by either IAq or CAq were placed in separate hive boxes according to queen source, and put in a temperature-controlled dark environmental room (32–34 °C, 30–40% relative humidity) [35]. Upon emergence, 50 workers from IAq or CAq were placed in Plexiglas cages (dimensions: 11.5 by 7.5 by 16.5 cm^3^). Sixteen cages were established for each trial: (2 pollen types (IAp or AZp) × 2 queen sources (IAq or CAq) × 4 replicate cages per treatment group).

Bees in cages were fed either AZp or IAp. The pollens were ground into powder with a coffee grinder (Mr. Coffee model 1DS77) and mixed thoroughly before feeding them to the bees. The ground pollen (10 g) was inserted into a plastic tube (2.2 cm in diameter) positioned on the side of each cage. CAq and IAp workers in the cages were fed either AZp or IAp. Each cage had a bottle with 30 mL of 50% sucrose solution and another with 30 mL of water. Pollen, sucrose and water were fed ad libitum. Sucrose solution was exchanged with fresh solution on day 4 and 7 to reduce mold growth. A piece of wax foundation was hung in the middle of each cage near the sugar solution and water vials and the bees clustered on it. Cages were maintained at a temperature of 32–34 °C.

Pollen consumption was measured by weighing the tube containing pollen as a single unit prior to feeding (initial weight), and after day 4 and 7 days to estimate consumption. On d-4, the tube was weighed to estimate pollen consumption, and replaced with a pre-weighed tube containing fresh pollen. On d-7, the tube was weighed and the amount of pollen consumed was added to the d-4 estimate to determine the amount of pollen consumed over 7 days.

Protein digestion, hemolymph protein concentration and HPG acini size in the spring and fall trials were estimated in the same five bees sampled from each cage on d-7. The bees were placed in separate vials, and stored at −20 °C until analysis. An additional 10 bees were sampled per cage, placed in separate vials and stored at −80 °C until analysis for expression of *vg* and *hex 70*, and *110*.

### 2.6. Estimating Pollen Protein Digestion

Protein concentration was measured in the hindgut contents of 7 day old bees fed spring or fall IAp or AZp using methods described in [36]. Briefly, hindguts were removed from the abdomens of five bees per cage (4 cages per treatment group). An incision into the hindgut was made and a 1 ul sample of the gut contents was taken. The contents were transferred to a 2 mL microcentrifuge tube containing 99-µL of phosphate buffered saline (PBS) with 1% EDTA-free Halt Protease Inhibitor Cocktail (Thermo Scientific). The gut contents from the five bees were pooled to generate a single sample for the cage. Samples were stored at −80 °C until analysis by a BCA protein assay (Thermo Scientific). Standard curves to estimate soluble protein concentration in the samples were prepared using bovine serum albumin. Protein absorbance was measured at 562 nm using a Biotek Synergy HT spectrophotometer. The proportion of digested pollen protein was estimated as: 1-(protein concentration in the hindgut contents/protein concentration in the pollen). Higher levels of protein in the hindgut indicated lower levels of protein digestion.

### 2.7. Hemolymph Protein Concentration

Hemolymph was drawn from bees by using a 20-μL pulled capillary tube inserted into the right lateral portion of the thorax near the wings. Additional hemolymph, if needed, was collected by inserting the same tube into the membrane between the abdominal tergites. Approximately 1–5 μL of hemolymph was collected per bee in the capillary tube, and then dispensed on to wax paper. A 1 μL sample of hemolymph per bee was transferred to a 2 mL microcentrifuge tube containing 99 μL of PBS with 1% EDTA-free Halt Protease Inhibitor Cocktail (Thermo Scientific). Hemolymph samples were stored at −20 °C until analysis. The samples were prepared for analysis of soluble protein by adding 100 μL of hemolymph to 900 μL of PBS with 1% Halt. Soluble protein concentration was analyzed using a BCA protein assay (Thermo Scientific), and quantified as described above for protein concentration in gut contents [36].

### 2.8. Measuring Hypopharyngeal Glands

HPG were measured in 7-day old bees fed IAp or AZp. Five bees were collected from each of the 16 cages (see above) established in spring and fall trials. The bees were stored at −80 °C until their HPG were measured using previously described techniques [35,37]. Briefly, HPG were removed from head capsules and placed into PBS (37 mM NaCl, 2.7 mM KCl, and 10 mM PO_4_, pH 7.4). The HPG were examined microscopically at 60 × magnification, and the area (mm^2^) of five randomly selected acini per bee was measured using the Leica Applications Suite v.3.8.0 software. Only acini with clear borders were measured. Acini areas were averaged within individuals and then among individuals from the same cage to obtain an estimate of HPG size for the cage.

### 2.9. Expression of vg, hex 70, and hex 110

The expression of three genes—vitellogenin (*vg*), hexamerin 70a (*hex70a*), and hexamerin 110 (*hex110*)—were measured in the fat bodies of 7d bees. Fat bodies were dissected and RNA extracted as described in [38]. Primers for *vg*, *hex70a*, *hex110*, and *actin* (Appendix A) were used to synthesize gene-specific first-strand cDNA from the total RNA (ThermoFisher RevertAid). These samples were subjected to an individual qRT-PCR reaction (SsoAdvanced Universal SYBR Green Supermix, BioRad) for each primer pair. The resultant Cq values were averaged across technical replicates. Expression of *vg*, *hex70a*, and *hex110* were normalized to *A. mellifera* actin. Relative expression estimates were obtained using the 2^−ΔΔCt^ method [38].

### 2.10. Statistical Analysis

All analyses were conducted using JMP (Cary, NC, USA) or Minitab (State College, PA, USA) statistical software. Average soluble protein and lipid concentrations in spring and fall IAp and AZp were compared using analysis of variance followed by Tukey’s or Fishers pairwise comparisons for post hoc analysis. Comparisons of amino and fatty acid concentrations between IAp and AZp (spring and fall) were made using analysis of variance. The effects of pollen type and queen source on pollen consumption, protein digestion, total protein consumed, hemolymph protein concentration, and HPG size were determined using a two-way analysis of variance using pollen and queen types as factors in the general linear model. Separate analyses were conducted for spring and fall trials. Whether there were seasonal effects on the parameters listed above within the same queen type when fed spring vs. fall pollens was determined using a general linear model with trial and pollen source along with the interaction term as factors in the model. An additional analysis was conducted using a multifactorial analysis of variance to compare responses to seasonal pollens between workers from the same queen type using data from spring and fall trials. Queen type, pollen type and trial were factors in the model. Gene expression (*hex 110*, *hex 70a* and *vg*) were compared between queens, pollen types and seasons using Kruskal–Wallis tests.

## 3. Results

### 3.1. Floral Composition of Pollens

Spring pollens from Arizona and Iowa differed in floral sources (Figure 1). Spring AZp was comprised exclusively of *Brassica* spp. Spring IAp was primarily *Trifolium* and *Mellilotus* spp. with small amounts of *Cryptotaenia* and *Schoenocrambe* spp. Fall AZp contained mostly *Ambrosia* and *Corethrogyne* spp. with small amounts of *Pectis* spp. Fall IAp was primarily *Trifolium* and *Symphotrichum* with small amounts of *Allium*, *Chamaecrista*, *Ambrosia* and *Solidago* spp.

### 3.2. Comparisons of Protein, Lipids, Amino and Fatty Acids in Spring and Fall Pollens

Pollens differed in soluble protein concentrations and amino acid composition depending on season and location where they were collected. Spring AZp had significantly higher soluble protein than spring IAp (Table 1). Protein concentrations were similar between fall AZp and IAp. Comparing across seasons, protein concentration in AZp was similar between spring and fall; spring IAp had lower protein concentrations than fall IAp. Amino acid concentrations were similar between spring AZp and IAp, and between fall AZp and IAp. Spring pollens had higher amino acid concentrations than fall pollens.

Within season comparisons between individual amino acids in spring AZp and IAp indicated similar concentrations of all essential amino acid except threonine, which was higher in IAp (F_3,20_ = 14.47, *p* < 0.0001) (Figure 2). Concentrations of the conditional amino acids: glycine (F_3,20_ = 4.6, *p* = 0.013), serine (F_3,20_ = 6.98, *p* = 0.002), and arginine (F_3,20_ = 46.48, *p* < 0.0001) were significantly higher in spring IAp than AZp. In fall pollens, essential (valine—F_3,20_ = 6.83, *p* = 0.002, leucine—F_3,20_ = 34.93, *p* < 0.0001, isoleucine—F_3,20_ = 110.43, *p* < 0.0001, threonine (see above), and phenylalanine—F_3,20_ = 62.03, *p* < 0.0001), conditional (serine (see above), proline—F_3,20_ = 9.93, *p* < 0.0001, and tyrosine—F_3,20_ = 7.55, *p* = 0.001), and nonessential amino acids (cysteine—F_3,20_ = 6.13, *p* = 0.004, alanine—F_3,20_ = 33.38, *p* < 0.0001 and aspartic acid—F_3,20_ = 15.70, *p* < 0.0001) were higher in IAp than AZp. Only cysteine was higher in fall AZp compared with IAp (F_3,20_ = 6.13, *p* = 0.004).

Comparing amino acid concentrations between spring and fall AZp and IAp indicated similar trends between pollens from the different locations. Spring pollens in both locations had similar or higher concentrations than fall pollens of the essential amino acids: valine, leucine, isoleucine, phenylalanine (see above), methionine (F_3,20_ = 4.55, *p* = 0.014) and lysine (F_3,20_ = 3.6, *p* = 0.031). The conditional amino acids, arginine (see above) and glutamine (F_3,20_ = 14.12, *p* < 0.0001) also were in higher concentrations in spring compared with fall IAp and AZp. Only histidine (F_3,20_ = 1.55, *p* = 0.232) and tryptophan (F_3,20_ = 0.04, *p* = 0.989) concentrations were similar between seasons at both locations.

Pollen also was analyzed for lipid and fatty acid composition. Lipid concentrations were similar between spring IAp and spring AZp, but higher in fall IAp than fall AZp (Table 1). Spring and fall IAp had similar lipid concentrations. AZp had higher lipid concentrations in the spring than the fall. Ratios of lipid to protein were higher in spring IAp than spring AZp. Fall pollens had similar protein to lipid ratios.

Total fatty acid concentrations ranged from 2.9% to 3.6% of total wet pollen mass. Spring AZp and IAp had higher concentrations of total fatty acids compared with fall pollens (Table 1). Total fatty acid concentrations in spring IAp and spring AZp were similar, but concentrations in fall IAp were significantly higher than fall AZp.

Fatty acids found in highest concentrations were palmitic, α-linolenic and linoleic acids (Figure 3). Spring IAp and AZp had similar concentrations of palmitic acid, but fall IAp had higher concentrations than fall AZp (F_3,20_ = 24.6, *p* < 0.0001). The essential fatty acids, α-linolenic (omega-3) and linoleic (omega-6) differed between IAp and AZp with IAp having higher concentrations of omega-3 (F_3,20_ = 102.28, *p* < 0.0001) and AZp having greater amounts of omega-6 (F_3,20_ = 101.56, *p* < 0.0001). Omega-6:3 ratios for spring AZp were 1.1:1 and 1.5:1 for fall. In IAp, ratios were 0.31:1 for spring and 0.23:1 for fall. Omega-6 concentrations were highest in spring AZp and lowest in fall IAp. Omega-3 concentrations were higher in spring compared with fall pollens; spring and fall IAp had higher concentrations of omega-3 than AZp.

Fatty acids found at lower concentrations in pollen samples also differed between spring and fall pollens and between IAp and AZp. Cerotic (F_3,20_ = 11.33, *p* < 0.0001) and arachidic acids (F_3,20_ = 41.76, *p* < 0.0001) were detected in the highest concentrations in AZp. Montanic, lauric, and myristic acids occurred at higher concentrations in spring and fall AZp compared with IAp (montanic—F_3,20_ = 11.87, *p* < 0.0001, lauric—F_3,20_ = 22.09, *p* < 0.0001, myristic—F_3,20_ = 103.5, *p* < 0.0001).

### 3.3. Seasonal Pollen Consumption, Protein Digestion, Protein Consumption, Hemolymph Protein Concentrations, and Hypopharyngeal Gland Size

#### 3.3.1. Do Workers Respond Differently to Pollen Mixes That Differ in Composition?

In the spring trial, the amount of pollen consumed was affected by queen and pollen type (Appendix A). More IAp was consumed than AZp by CAq and IAq (Appendix A). CAq and IAq digested less of the protein in IAp than AZp (Appendix A). We estimated the total protein consumed from each pollen type by the bees after 7 days by: μg of protein per mg of pollen × mg of pollen consumed × proportion of pollen protein digested. Workers from both queen types ingested significantly more protein from AZp than IAp despite consuming more IAp than AZp (Figure 4). Even though protein consumption was higher when bees consumed AZp, hemolymph protein concentrations were higher and HPG acini were larger in bees consuming IAp (Figure 4).

In the fall trial, pollen and total protein consumption were affected by queen but not pollen type (Appendix A). CAq and IAq consumed similar amounts of fall AZp and IAp (Appendix A). However, CAq consumed more pollen than IAq. Hemolymph protein levels were affected only by pollen type; levels were higher in CAq and IAq that consumed IAp. HPG sizes were unaffected by pollen or queen type and were similar for CAq and IAq bees consuming either pollen type.

#### 3.3.2. Do Workers Respond Differently to Pollen Based on Season?

The analysis comparing responses to spring vs. fall pollens from CAq and IAq indicated seasonal effects on all response variables (Appendix A). The bees responded differently to spring and fall pollens in that more pollen was consumed in spring than fall, but more of the protein in fall pollens was digested. Higher protein digestion rates resulted in fall bees consuming more protein than spring bees (Figure 4). Though bees consumed more protein from fall than spring pollens, hemolymph protein concentrations were higher and HPG were either similar to (IAq) or larger (CAq) in spring bees (Figure 4).

#### 3.3.3. Do Responses to Season and Pollen Sources Differ between Queen Lines?

There were differences between queen types consuming seasonal AZp and IAp for all response variables (pollen and protein consumption, digestion, and HPG) except hemolymph protein levels (Appendix A). The amount of pollen consumed was affected by pollen and queen type and the interaction between them. Bees consumed more spring than fall pollen, more IAp than AZp, and CAq consumed more pollen than IAq particularly in the spring (Appendix A). Protein digestion was affected by queen type and season, and there were significant interactions between pollen type and season and queen type and season. CAq digested similar amounts of AZp and IAp, while IAq digested more AZp. More pollen protein was digested from fall pollens, and more protein was consumed by CAq than IAq in the fall. Hemolymph protein concentrations also were significantly affected by season and pollen type. Concentrations of protein in the hemolymph were similar between the queen types, and higher in spring compared with fall bees especially if bees consumed IAp. HPG were affected by queen, pollen type and season. In general, HPG were smaller in IAq than CAq and in the fall compared with spring particularly when bees consumed IAp. Though fall bees consumed more protein from fall than spring pollens, HPG were smaller in the fall bees.

#### 3.3.4. Expression of *hex 70a*, *hex 110* and *vg*

Both CAq and IAq had similar expression levels of genes associated with nutrient storage (*hex 70a 110*, or *vg)* when consuming either AZp or IAp in the spring or fall trials (Appendix A). There were however, differences in CAq workers consuming spring vs. fall pollen with significantly higher expression of *vg* (AZp and IAp) and *hex 70a* (AZp) when they consumed spring pollen. There were no differences in gene expression based on season in IAq consuming either pollen type. Comparisons between CAq and IAq consuming the same pollen indicated significantly higher expression of *hex 70a*, *110* and *vg* when CAq consumed IAp. Expression levels of *vg* in CAq also were higher than IAq when consuming AZp. There were no differences between CAq and IAq in expression of the genes we examined when they consumed either fall AZp or IAp.

## 4. Discussion

Our study addressed two questions related to nutrients in seasonal pollens and how they are utilized by honey bees. The first is whether nutrients differ based on season and floral composition between locations. The second question is whether bees respond to pollen differently depending on season and region where their queens were sourced. There were differences in nutrient content between spring and fall pollens collected by bees in Arizona and Iowa. Nutrients also differed between locations even when pollen was collected during the same season. We found effects from queen type in responses to the parameters we measured that may reflect genetic differences between CAq and IAq, We also found differences in seasonal responses to pollens by bees from both queen lines. Notably, bees consumed more protein from fall than spring pollens, but fall bees had smaller HPG.

The pollens comprising spring and fall AZp and IAp were collected by colonies and were products of the apiary locations and the conditions during the year when the pollen was collected. Spring AZp was comprised of *Brassica* spp. and IAp was primarily *Trifolium spp*. Other species may have been present in our samples, but at concentrations that were too low to detect. The limited pollen diversity may have been because the Arizona and Iowa apiaries were on research farms rather than in undisturbed ecosystems. Weather conditions also influence the number and diversity of species in the bloom. Plant sources represented in the spring and fall pollen mixtures from Arizona and Iowa did not have genera in common with the exception of *Ambrosia* spp. in the fall. Although we did not use California pollens due to logistical constraints, AZp was comprised of genera from plants grown in the region where CAq originated. For example, spring AZp was predominantly mustards (*Brassica* spp.) that grow in regions of California where CAq were sourced [39]. Fall AZp was predominantly *Ambrosia* and *Corethrogyne* spp. that also are found in regions of California where CAq originated [40]. Fall IAp were primarily clovers, and with the exception of a small amount of *Ambrosia* spp., differed from AZp.

Though pollen identification might have missed some pollen species present in low concentrations, those pollens still would have contributed to the nutritional profile of the location and season. There were differences in amino acid concentrations between spring and fall pollens collected from the same region. In agreement with previous reports [19], all essential amino acids were in equivalent or higher concentrations in spring compared with fall pollens. Essential amino acids that support brood rearing and larval development (e.g., valine [41], isoleucine [42], leucine [43], and lysine) were in higher concentrations in spring compared with fall pollens from both regions. Conditional amino acids also were in higher concentrations in spring pollens as previously reported [19]. These amino acids (e.g., glutamine) play a role as nutritional drivers for the upregulation of genes associated with the mTOR signaling pathway and assembly of major royal jelly proteins (serine) [44,45,46]. Our data suggest that the nutritional profile of spring pollens collected by honey bees may be uniquely suited to support a colony’s brood-rearing activities. Additional studies of spring and fall pollens collected by colonies in different regions and comprised of other species are needed however, to test the consistency of this trend.

Comparing seasonal pollens between geographic regions, we expected spring AZp and IAp to be similar in macronutrients, amino acids, and fatty acids since colonies in both regions are expanding during this time of year. Since spring pollens in our study were comprised primarily of *Brassica* spp. (AZp) and *Trifolium* spp., (IAp) our comparisons are in large part, between these two sources and between a monofloral (AZp) and polyfloral (IAp) mix. While spring AZp had higher protein concentrations than IAp, amino acid composition between the pollens was similar except for threonine which was higher in IAp. Nutrients in fall pollens support overwintering strategies, and since these strategies differ between the regions where the pollen was collected, we expected the nutrient composition of AZp and IAp to differ. We found that fall AZp and IAp had similar protein concentrations, but amino acid composition differed. Amino acids that might be beneficial to colony survival during winter confinement were in higher concentrations in IAp than AZp. These included isoleucine and proline. Higher levels of isoleucine could support brood rearing late in the winter while the colony is confined since it is a brood pheromone carrier [42]. Proline was in higher concentrations in fall IAp than AZp. Higher concentrations of this amino acid could be beneficial to colonies confined during the winter because proline serves as a carbon shuttling molecule between lipid reserves in the fat body and flight muscles that are used to generate heat and thermoregulate the winter cluster [47,48]. A finding that was similar to a previous study, was that both CAq and IAq consumed more spring than fall pollen supporting colony expansion. However, more protein was digested from fall pollens so that fall bees consumed more protein than spring bees possibly in preparation for periods of confinement [19].

Protein and amino acid concentrations often are used to evaluate the nutritional quality of pollen [49,50]. Recent studies however, have shown that lipid and fatty acid levels also are important components of nutritional quality as they support brood development and adult longevity, both of which are cornerstones to colony growth [51,52]. Lipid levels were similar between spring AZp and IAp. However, fall IAp lipid levels were significantly higher than fall AZp. Higher lipid concentrations in fall IAp might better support colonies that are confined for the winter, and rely on nutrients stored in the fat bodies of workers that comprise the winter cluster [53].

In our study, lipid percentages and total fatty acid concentrations were within previously published ranges for bee collected pollen mixtures [54,55]. Palmitic, omega-3 and omega-6 were the predominant fatty acids found in AZp and IAp. These results are similar to previous reports of pollens collected by honey bees [19,54,56]. However, spring IAp (primarily *Trifolium* spp.) had significantly higher levels of omega-3 than AZp (*Brassica* spp.) and lower omega-6:3 ratios. Omega-3 is an essential fatty acid [57,58], and affects HPG development [52,59]. The higher omega-3 levels and lower omega-6:3 ratios in IAp may explain why CAq and IAq developed larger HPG when consuming IAp. This occurred despite consuming significantly less protein from IAp than AZp. Similar results were reported by [52], and suggest that HPG development might be more closely tied to lipid (specifically omega-3) levels than to protein. This finding also indicates that *Trifolium* spp. could be an important pollen source for colony growth due to omega-3 levels and should be included in pollinator plantings.

Since pollen provides bees with nutrients required for colony growth and maintenance, pollens with higher nutrient density should be collected and consumed at higher rates [60,61], though this does not consistently occur [59]. In our study, lipid levels were similar between spring AZp and IAp, but AZp had a higher protein concentration. Despite the higher protein levels in AZp, CAq and IAq consumed more IAp. Greater consumption of IAp might be due to the higher diversity of pollen types represented in IAp. Alternatively, higher consumption of IAp might be related to the relative ability of bees to digest the protein the pollen provided. Less of the protein in IAp was digested than AZp. There might be an inverse relationship between consumption and digestion at the level of individual bees, and perhaps the colony level that gives the appearance that one pollen type is more attractive to bees than another. The inconsistencies in selection of pollens might be more nuanced than just nutrient content, and include factors like the ability of bees to access the nutrients the pollen provides.

CAq and IAq responded differently when fed the same pollen. The differences might be because the queens were commercially produced from different subspecies lineages and open mated in different geographic areas. CAq were Italian (Ligustica) queens that are generally characterized as having rapid colony growth in the spring in response to food supply [62]. In our study, CAq consumed more pollen and had larger HPG in the spring than IAq. Gene expression levels also differed between spring CAq and IAq with higher expression levels for *hex70a*, *110* and *vg* occurring in CAq. These traits could support the characteristic rapid colony growth in the spring associated with Italian stock. IAq were Carniolan, a subspecies characterized as having decreased brood rearing and reduced food consumption in the fall, and good overwintering capabilities [63]. These traits were evident in the fall trial as IAq consumed significantly less pollen and protein than CAq. IAq also differed from CAq in that they did not have seasonal differences in HPG size or gene expression levels. HPG size in IAq being similar between trials was unexpected since these organs generally are smaller in bees in the fall as brood rearing declines. This trait might be variable with respect to subspecies and queen line. HPG produce brood food, but also can contain elevated protein levels and serve as storage organs similar to the fat body in winter bees [14,64]. Our study indicates that the relationship between HPG size and season might be influenced by the subspecies and breeding conditions of the queens. However, further studies are needed to determine if the relationships among season, pollen consumption and HPG size vary based on queen line and location where the queens are mated.

There were differences in responses to pollen feeding between queen types and seasons, but our results were from a cage study and should be interpreted cautiously. Cage studies are a useful tool for hypothesis testing because colony level variation is controlled. However, colony level studies are needed because the effects of brood and young adult bees on physiology, gene expression, aging and life history of worker bees cannot be assessed in cages [44,65] Interestingly, we detected seasonal differences in the responses of AZq and IAq particularly in protein consumption and its relationship to HPG size between spring and fall trials, even though bees emerged and then were kept in cages in an environmental room. Bees consumed more protein in the fall trial yet had HPG that were similar in size or smaller than in the spring. The responses may be in part because of differences in nutrient composition between spring and fall pollens. The timing of bee emergence also might have contributed to seasonal effects since bees reared in the summer differ from those reared in the fall [13,66,67]. Our results suggest that the blueprint for how nutrients are used in summer and winter bees might be laid down in the larval stage. Others have found changes in adult honey bees due to larval nutrition [68,69], and characterized the responses as anticipatory mechanisms to program adults to better cope with changing environmental conditions (e.g., summer vs. winter colony conditions) [69]. Additional studies are needed to test this hypothesis, and determine if gene expression in larvae and subsequent behavioral responses as adults reared on seasonal pollens during periods of colony expansion differ from those in larvae reared during colony contraction and preparation for overwintering.

Our study demonstrated the flexibility that honey bees show in their use of pollen from different plant species (e.g., CAq developing large HPG while feeding on IAp). This ability contributes to honey bees’ capacity to migrate and establish nests in a wide range of environments, and to build populations after being moved to new food sources for pollination and honey production. While bees could utilize nutrients in the pollens regardless of where they originated, many of the factors we measured were influenced by the season when the pollen was collected. The seasonal responses might be due to a combination of differences in the nutritional composition of spring and fall pollens, and the bees’ genetic structure that influenced how the nutrients were used to support activities of the yearly colony cycle.

## 5. Conclusions

Pollens collected from two geographic regions differed in nutrient composition depending on season and location. There were seasonal effects in responses to every parameter we measured that may have occurred from a combination of genetic differences between the bees and the nutrient composition of spring and fall pollens.

## Figures and Tables

**Figure 1 insects-12-00235-f001:**
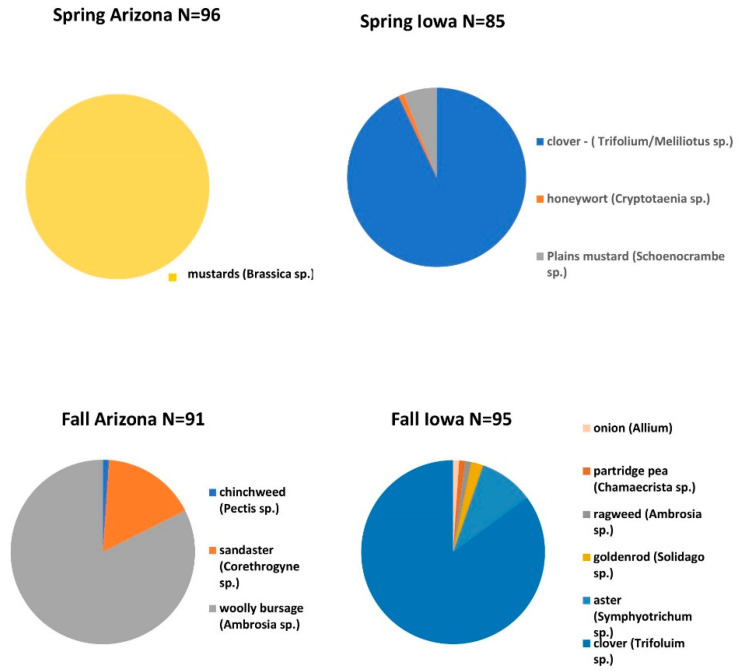
Proportions of pollens from plant species collected in pollen traps attached to honey bee colonies in Arizona (Pima County) or Iowa (spring—Story County, Fall—Story and Polk Counties). Pollen was collected in the spring and fall. Identification of pollen types was conducted using ITS gene sequencing [28,30]. For each seasonal pollen mix, ‘N’ represents the number of quality ITS DNA sequences obtained from each library.

**Figure 2 insects-12-00235-f002:**
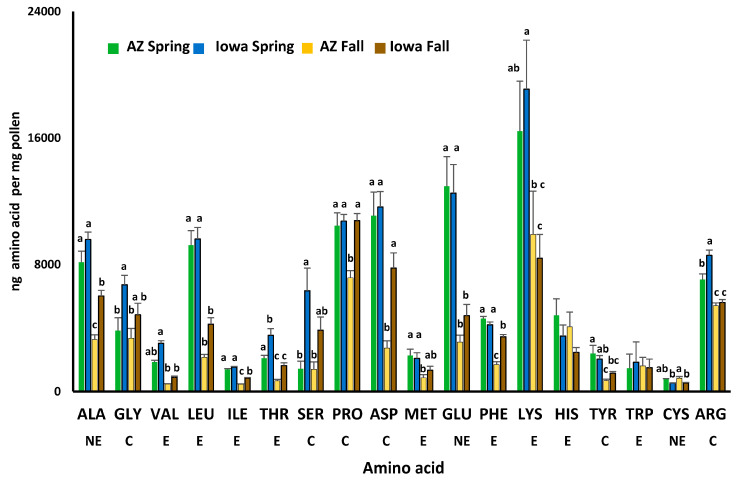
Essential (E), conditional (C), and nonessential (NE) amino acids in polyfloral mixtures of spring and fall pollens collected in Arizona (AZ) (Pima County, AZ, USA) and Iowa (Spring-Story County, Fall-Story and Polk Counties, IA, USA). Separate F-tests were conducted for each amino acid to compare mean concentrations among pollen types. Means for each amino acid followed by the same letter are not significantly different at *p* = 0.05. Histidine (HIS) and tryptophan (TRP) had no significant differences among the means.

**Figure 3 insects-12-00235-f003:**
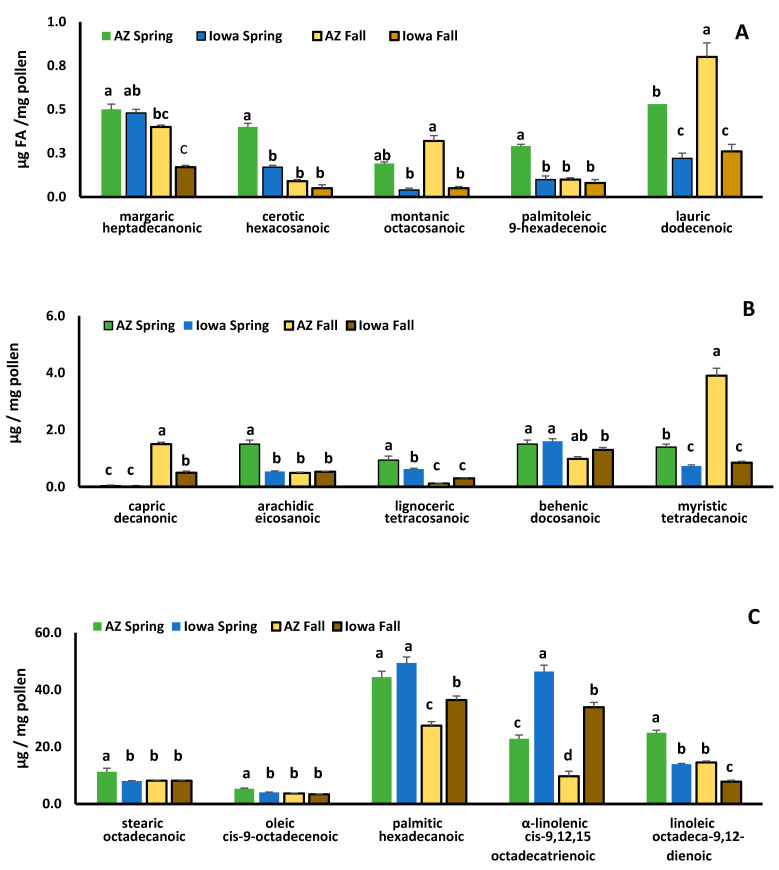
Fatty acid concentrations in polyfloral mixtures of spring and fall pollens collected in Arizona (AZ) (Pima County, AZ, USA) and Iowa (Spring-Story County, Fall-Story and Polk Counties, IA, USA). Fatty acids found in lowest concentrations are shown in (**A**), medium concentrations in (**B**), and highest concentrations in (**C**). Separate F-tests were conducted for each fatty acid to compare mean concentrations among pollen types. Means for each fatty acid followed by the same letter are not significantly different at *p* = 0.05.

**Figure 4 insects-12-00235-f004:**
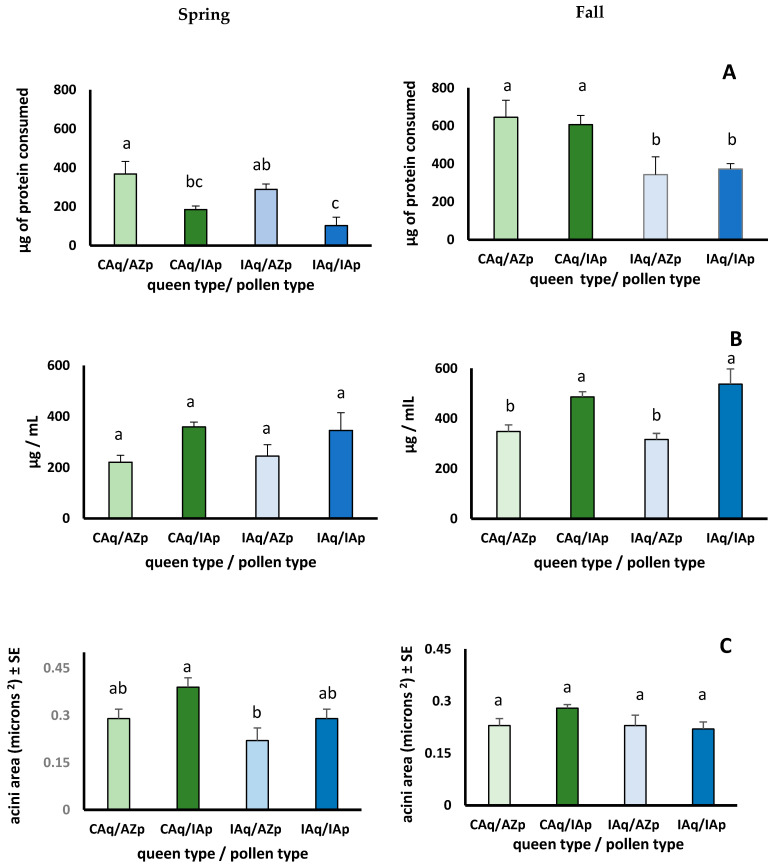
Average (±SE) amount of protein consumed (**A**) hemolymph protein concentration (**B**) and size of hypopharyngeal gland acini (**C**) in 7-day old honey bees that are offspring of queens from California (CAq) or Iowa (IAq) that were fed pollen collected in Arizona (AZp) or Iowa (IAp) in either the spring of the fall. Comparisons among queen/pollen type within the season were made using analysis of variance (protein consumption—spring: F_3,12_ = 7.73, *p* = 0.004, fall: F_3,12_ = 4.88, *p* = 0.019, hemolymph protein concentration—spring: F_3,12_ = 8.24, *p* = 0.003, fall: F_3,12_ = 2.39, *p* = 0.119, acini area—spring: F_3,12_ = 3.93, *p* = 0.036, fall: F_3,12_ = 1.40, *p* = 0.291). Means with the same letter within the same plot are not significantly different at the *p* = 0.05 level as determined by analysis of variance followed by Fisher pairwise comparisons.

**Table 1 insects-12-00235-t001:** Average total amino and fatty acids in pollens collected during the spring and fall from Iowa (Spring-Story County, fall-Story and Polk Counties, IA, USA) and Arizona (Pima County, AZ, USA). All concentrations are expressed as average micrograms (µg) or nanograms (ng) per milligram (mg) of pollen ± standard error. Means followed by the same letter are not significantly different at *p* = 0.05 as determined by Fisher pairwise comparisons.

Concentrations	Arizona Spring	Iowa Spring	Arizona Fall	Iowa Fall	F *	*p*
total soluble protein (µg/mg)	353.7 ± 7.5 a	275.8 ± 14.2 b	368.8 ± 29.1 a	379.4 ± 29.9 a	5.28	0.015
total amino acids (ng/mg)	102.4 ± 9.0 a	116.0 ± 0.1 a	49.7 ± 6.0 b	70.0 ± 4.8 b	17.1	<0.0001
total lipids (µg/mg)	127.2 ± 8.0 ab	157.7 ± 21.8 a	87.9 ± 12.8 b	148.3 ± 12.5 a	4.45	0.025
total fatty acids (µg/mg)	33.5 ± 1.7 a	36.6 ± 1.4 a	20.9 ± 1.0 c	27.1 ± 1.1 b	27.4	<0.0001
Protein to lipid ratio	1:0.36 b	1:0.57 a	1:0.24 b	1:0.39 ab	8.36	0.003

* d.f. for average total soluble protein and total lipid is 3, 12, d.f for average total amino and fatty acid is 3, 20.

## Data Availability

Data is available upon request from the corresponding author.

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
