# Peer review of "The Importance of Time and Place: Nutrient Composition and Utilization of Seasonal Pollens by European Honey Bees (Apis mellifera L.)"

_insects, 2021, doi:10.3390/insects12030235_

Round 1

Reviewer 1 Report

The manuscript titled »The importance of time and place: Nutrient composition and utilization of seasonal pollens by European honey bees (Apis mellifera L.)« discusses the differences in nutrient composition of pollen between different regions and seasons as well as the physiological responses of honey bees of different origin. The presented data are interesting, however few things need to be changed, mostly from conceptual aspect before publication.

I would like to address the concept of »local« in two ways. First, the vegetation list for Tucson AZ area contains about 20 Brassicacea taxa, but also many other that flower at the same time. Fig 1 shows 100% Brassica spp. in AZp and this result seems to hint at the strong agricultural source nearby, attracting the majority of bees (could authors confirm this?). If so, I would claim that such pollen could hardly be considered as local and the relevant sections of the manuscript should be changed accordingly. Same goes for clover in Iowa. Second, the »local« in in terms of evolutionary adaptations to food sources – in the vegetation of native ranges of both subspecies used (CE/Mediterranean Europe) neither the Brassicaceae neither clovers are dominant, except in agriculture environment. Outside such environment is a rich diversity of other flowering plants available at the same time as Brassicaceae, also orchards, wild Prunus trees etc. Is it justified to expect adaptation to »local« if in native environment local sources are different and much more diverse? If there was coevolution between the bees and the sources, it was between native bees and native plants. I would ask authors to provide some comments along these lines in the discussion.

L532 – 552: difference between Italian and Carniolan: both are remnant of the ice age, and while it is true that Am carnica is more continental, one can find Carniolan also at the Mediterranean coast in Croatia, right across the Adriatic Sea from Italian peninsula (for confirmation of distribution see nice albeit local paper from Bozic et al 2016; also Puskadija et al 2020). The climatic conditions and vegetation are similar, and one can expect reasonable plasticity within the native range of Carniolan honeybee, possible more than in Am ligustica since it covers more diverse environments. Simply concluding that differences in gene expression or in HPG are to be ascribed to certain race/subspecies could be wrong. I would ask authors to reform their conclusions.

Pollen analysis with ITS. My understanding of the ITS method is that it can easily miss the under-represented pollen species. If so, this could also explain low variation in pollen diversity. In my experience of palynological analysis of pollen samples there are always many species present, at least in minor shares. I kind of miss category »other« in pie charts of figure one, with some table stating these other taxa.

The low diversity of pollen also brings me to the »time« concept of the MS: I would say it is difficult to link results (AA, FA, etc…) to season having only a single species in the analysis: I would claim that authors actually are presenting the results for Brassica spp. and not for the spring pollen.   

Next, I would like authors to comment if the possible reason for the preference for IAp stem from higher pollen diversity? A single species pollen is poor and boring diet. A cross-comparison experiment might be useful to show preference: to feed spring bees with fall AZp and fall IAp alongside the spring AZp and spring IAp (and repeat the same with fall bees).

Simple summary:

L18: space too many

Abstract:

L34: space too many

Introduction

L55: space too many

L61: space too many

L81: space too many

Materials and methods

L138: not clear where USDA-ARS Carl Hayden BRC facility is located (I assume that it is AZ from later narrative.

L142: non-US citizens might not know what AZ stands for, same goes for CA and IA earlier; please replace with full name.

L142: does it matter why the queens from AZ were not used? You are passing same information in L138 by stating that experiment was done by European honey bees.

L157 – 163: how did you ensure coverage of least represented species? Did it matter at all?

L167, L171: pollen was pooled by location/season. How were then samples for soluble protein (3x) or AA (6x) taken? Out of the pooled sample by three random »grabs«? Was pooled pollen homogenized before sampling (to get »balanced« composition) or were there any similar approaches employed?

Results

Fig. 1. top right (honeywort not honewort)

Discussion

L470-471: predominantly mustards (spring AZp) or entirely mustards (Fig 1, results, L291)?

Supplement

Sup FigS1: a lonely letter »A« on p.2

Sup tables: ensure that the individual table fits on single page with all the comments below

Author Response

Below are my responses to each of the comments made by the Reviewer:

Reviewer-1

The manuscript titled »The importance of time and place: Nutrient composition and utilization of seasonal pollens by European honey bees (Apis mellifera L.)« discusses the differences in nutrient composition of pollen between different regions and seasons as well as the physiological responses of honey bees of different origin. The presented data are interesting, however few things need to be changed, mostly from conceptual aspect before publication.

I would like to address the concept of »local« in two ways. First, the vegetation list for Tucson AZ area contains about 20 Brassicacea taxa, but also many other that flower at the same time. Fig 1 shows 100% Brassica spp. in AZp and this result seems to hint at the strong agricultural source nearby, attracting the majority of bees (could authors confirm this?).

Response – AZp was collected at a University of Arizona Research Farm so it is correct that the agricultural setting (L172-173) may have excluded some of the native plants that bloom from February to April in the Tucson area. While a cultivated stand of Brassica did not exist, there were mustard plants in bloom in the borders of the agricultural fields and open areas. However, the farm borders the Rillito River and spring wildflowers can grow there. This area was within the foraging range of our apiary. The germination, growth and flowering of spring ephemerals is dependent on winter rainfall that can limit the size of stands and their attractiveness to a colony’s foraging population. If plants were blooming in large enough densities, I would have expected them to be foraged on by bees and been detected in our pollen traps. Still, I agree with the Reviewer that having Brassica as the sole pollen species in out AZp might not be representative of pollen diversity that can occur in other locations and years. This point was made in the Discussion: (‘L538-540) ‘The pollens comprising spring and fall AZp and IAp were  collected by colonies and were products of the apiary locations  and the conditions during the year when the pollen was collected. Spring AZp was comprised of Brassica spp. and IAp was primarily Trifolium spp. Other species may have been present in our samples, but at concentrations that were too low to detect. The limited pollen diversity may have been because the Arizona and Iowa apiaries were on research farms rather than in undisturbed ecosystems. Weather conditions also influence the number and diversity of species in the bloom. Plant sources represented in the spring and fall pollen mixtures from Arizona and Iowa did not have genera in common with the exception of Ambrosia spp. in the fall. Although we did not use California pollens due to logistical constraints, AZp was comprised of genera from plants grown in the region where CAq originated. For example, spring AZp was predominantly mustards (Brassica spp.) that grow in regions of California where CAq were sourced [39]. Fall AZp was predominantly Ambrosia and Corethrogyne spp. that also are found in regions of California where CAq originated [40]. Fall IAp were primarily clovers, and with the exception of a small amount of Ambrosia spp., differed from AZp. (L538-553).

If so, I would claim that such pollen could hardly be considered as local and the relevant sections of the manuscript should be changed accordingly. Same goes for clover in Iowa. Second, the »local« in in terms of evolutionary adaptations to food sources – in the vegetation of native ranges of both subspecies used (CE/Mediterranean Europe) neither the Brassicaceae neither clovers are dominant, except in agriculture environment. Outside such environment is a rich diversity of other flowering plants available at the same time as Brassicaceae, also orchards, wild Prunus trees etc. Is it justified to expect adaptation to »local« if in native environment local sources are different and much more diverse? If there was coevolution between the bees and the sources, it was between native bees and native plants. I would ask authors to provide some comments along these lines in the discussion.

Response – The reviewer makes a good point. Coevolution between bees and pollen sources would be evident in native bees. Based on the comments of this reviewer and Reviewer-2 who makes similar arguments, any reference to local and non-local pollens and adaptations have been removed from the manuscript. The comparisons that are made are the same, but left at CAq feeding on AZp and IAp, and IAq feeding on AZp and IAp. The paper focuses on response of two different queen types feeding on pollen from two different geographic regions so that the species composition would be different between them, during two different seasons.

L532 – 552: difference between Italian and Carniolan: both are remnant of the ice age, and while it is true that Am carnica is more continental, one can find Carniolan also at the Mediterranean coast in Croatia, right across the Adriatic Sea from Italian peninsula (for confirmation of distribution see nice albeit local paper from Bozic et al 2016; also Puskadija et al 2020). The climatic conditions and vegetation are similar, and one can expect reasonable plasticity within the native range of Carniolan honeybee, possible more than in Am ligustica since it covers more diverse environments. Simply concluding that differences in gene expression or in HPG are to be ascribed to certain race/subspecies could be wrong. I would ask authors to reform their conclusions.

Response – While the point made by the reviewer is valid, we obtained our queens from breeders that maintain their genetic lines, as best they can, by grafting and open mating their queens in isolated apiaries. Though the differences we saw in responses between queen lines may not be solely from their subspecies and date back to their origins in the Ice Age, the differences can be ascribed to the lines selected by the breeders and the areas where they were open mated and that these differed between CAq and IAq. The paragraph was rewritten to clarify the point that the differences between CAq and IAq could be the product of how they were bred rather than solely because of subspecies (L649-669)

Pollen analysis with ITS. My understanding of the ITS method is that it can easily miss the under-represented pollen species. If so, this could also explain low variation in pollen diversity. In my experience of palynological analysis of pollen samples there are always many species present, at least in minor shares. I kind of miss category »other« in pie charts of figure one, with some table stating these other taxa.

 RESPONSE - Yes, we agree that ITS sequencing can underestimate the number of rare taxa in the samples compared to other methods, like light microscopy (Corby-Harris et al. 2018, Hawkins et al. 2015). However, ITS sequencing is sometimes better at detecting these rarer taxa (Smart et al. 2017). Overall, the two methods tend to qualitatively agree (Richardson et al. 2015). With just one method used to estimate taxon identity and richness, our estimates may not completely reflect the true plant species richness in the landscape. However, this should not be a problem for our study because the goal of the ITS sequencing was to simply show that species and mixtures of spring and fall pollens from Iowa and Arizona are different. 

The low diversity of pollen also brings me to the »time« concept of the MS: I would say it is difficult to link results (AA, FA, etc…) to season having only a single species in the analysis: I would claim that authors actually are presenting the results for Brassica spp. and not for the spring pollen.   

RESPONSE – The reviewer makes a good point, and this is a limitation of our study. The pollen the bees collected was predominantly Brassica spp. If our colonies were in another area of southern Arizona, the pollen composition probably would have differed. Based on the Reviewer’s comment, we have added the following information to the Discussion: ‘Since spring pollens in our study were comprised primarily of Brassica spp. (AZp) and Trifolium spp., (IAp) our comparisons are in large part, between these two sources. While spring AZp had higher protein concentrations than IAp, amino acid composition between the pollens was similar except for threonine which was higher in IAp.’ (L568-574). However, spring IAp showed a similar trend to AZp where spring pollen had higher total amino and fatty acids than fall IAp. The following sentence was added to the Discussion to make this point, ‘Additional studies of spring and fall pollens from different species collected by honey bee colonies are needed however, to test the consistency of this trend.‘ (L-567-569). An additional point was added in response to the Reviewer’s comment to the paragraph about the role of higher Omega-3 levels in the spring IAp compared to AZp, ‘Similar results were reported by [52], and suggest that HPG development might be more closely tied to lipid levels than to protein. This finding also indicates that Trifolium spp. could be an important pollen source for colony growth and should be included in pollinator plantings.’ (L611-613).

Next, I would like authors to comment if the possible reason for the preference for IAp stem from higher pollen diversity? A single species pollen is poor and boring diet. A cross-comparison experiment might be useful to show preference: to feed spring bees with fall AZp and fall IAp alongside the spring AZp and spring IAp (and repeat the same with fall bees).

RESPONSE – IAp may have been consumed at a higher rate than AZp in the spring trial because of greater pollen diversity, and this point was added to the Discussion; ‘Greater consumption of IAp might be due to the higher diversity of pollen types represented in IAp. Alternatively, higher consumption of IAp might be related to the ability of bees to digest the protein the pollen provided.’ (L619-621). However, I do not agree that a single species is necessarily a poor and boring diet so bees consume less of it. Honey bees collect and consume large amounts of single pollens, e.g., almond pollen, when colonies are pollinating crops. The experiment suggested by the reviewer was conducted and published (DeGrandi-Hoffman et al 2018) where spring and fall bees were fed spring and fall pollens collected in Arizona. The fall pollen had greater diversity than the spring pollen but spring bees still consumed more of the spring pollen.

Simple summary:

L18: space too many

 RESPONSE: The extra space was deleted (L18).

Abstract:

L34: space too many

RESPONSE – The Abstract was rewritten to remove points about local and non-local pollens, and the extra spaces were corrected.

 Introduction

L55: space too many

RESPONSE The extra space was deleted (L55)

L61: space too many

RESPONSE The extra space was deleted (L63)

L81: space too many

RESPONSE The extra space was deleted (L82)

Materials and methods

L138: not clear where USDA-ARS Carl Hayden BRC facility is located (I assume that it is AZ from later narrative.

Response – the address of the Bee Center was corrected to: ‘The study was conducted as a common garden experiment at the USDA-ARS Carl Hayden Bee Research Center in Tucson, Arizona, USA in 2016 with European honey bees.’ (L172-173)

L142: non-US citizens might not know what AZ stands for, same goes for CA and IA earlier; please replace with full name.

Response: The names of the states were replaced with full names when referred to throughout the manuscript. (e.g.,L-173,174,176, 177). Changes also were made in the Table and Figure captions.

L142: does it matter why the queens from AZ were not used? You are passing same information in L138 by stating that experiment was done by European honey bees.

RESPONSE: I think it is important to let the reader know why we did not use queens that were sourced and open mated in Arizona but instead used California queens. I agree that we already stated that the study was conducted with European bees so the sentence was changed to, ‘We did not use queens from Arizona since the resident population is Africanized [27].’ L178-179

L157 – 163: how did you ensure coverage of least represented species? Did it matter at all?

RESPONSE: We used pollen collected by colonies. Because of the foraging behavior of colonies, the species that had the most open flowers would predominate in our pollen samples. Floral species with less open flowers may not have been foraged with sufficient frequency to be detected in our pollen samples. That being said, if those pollens were in the pollen samples, they would have contributed to the nutritional composition of the pollen mixes. This point was made in the Discussion (L554-556).

L167, L171: pollen was pooled by location/season. How were then samples for soluble protein (3x) or AA (6x) taken? Out of the pooled sample by three random »grabs«? Was pooled pollen homogenized before sampling (to get »balanced« composition) or were there any similar approaches employed?

RESPONSE: A more detailed description of how we mixed the pollen collections from the different colonies over time was added to the Materials and Methods: ‘Pollen for all trials was removed weekly from traps and pooled in 2-gallon plastic Ziplock bags. The pollen was mixed by shaking and rolling the bag. A random sample from each weekly bag was taken and pooled into a new 2-gallon plastic Ziplock bag. The pollen in the bag was mixed as described above. A one liter sample of the pollen was removed from the bag for feeding the bees in the cages (see below). The procedure was repeated for fall AZp and for pollen collected in Iowa.’ L203-208.

 Results

Fig. 1. top right (honeywort not honewort)

Response- I could not correct the legend in the figure embedded in the manuscript. I have included a corrected Figure 1 with the revised submission.

 Discussion

L470-471: predominantly mustards (spring AZp) or entirely mustards (Fig 1, results, L291)?

RESPONSE: Predominantly is the correct term since other pollens may have been in the mix, but at levels too low to detect.

Supplement check this and correct

Sup FigS1: a lonely letter »A« on p.2

RESPONSE: I could not access Fig. S1 to correct it. I have uploaded a new Fig. S1 with the ‘A’ removed.

Sup tables: ensure that the individual table fits on single page with all the comments below

RESPONSE: I am not sure what the reviewer is referring to here.

Reviewer 2 Report

Please see comments in attached word doc. 

Author Response

Below are my responses to each of the comments by Reviewer-2:

Reviewer-2

Insects Review

The importance of time and place: Nutrient composition and utilization of seasonal pollens by European honey bee (Apis mellifera L.).

This paper is important because it looks at the variation in pollen sources and both temporally and geographically, and it also evaluates the effect of that variation on bees. Overall, the language is very clear, and the writing in general is very good. It was in a very professional and readable format.

The link to physiological changes from different types of pollen is incredibly useful and there is a huge gap in the literature for these types of studies. Overall, the approach - the field, lab, and statistical analysis are useful for assessing whether different types of pollen cause different health effects. The methods match the title well. While the experimental design and writing are good, the interpretation of the results are highly problematic. The conclusions drawn from this work extend waaay beyond the data presented. The language of ‘local’ and ‘adaptation’ are simply not supported either by the design of this project, or the literature, and the statements made to this effect and are distracting from the basic science that is presented. The extrapolation of the results beyond the bounds of this study are just too great to be published as is, and the analysis really makes the whole publication confusing. The paper has SUPER interesting results about how bees use protein differently - between times of year, and with different genetics. This as a stand alone result is useful to the body of literature. However, when this is couched as being representative of local adaptation or that these pollens represent regional variability, the work becomes distracting, and frankly misleading.

RESPONSE: The Reviewer has convinced me that extending our findings into evidence of local adaptation is perhaps going too far. I have removed all references to our study being focused on local and non-local pollens and local adaptation. I stuck with the direct comparisons we made that included queen lines, pollen types, and seasons.

This paper would be so much better if it just was written as if it were bees from two different queens/areas, fed pollen from two different areas, at two different times of year. It is still important, because is starts to outline the bounds of the variation that we see, and addresses a huge data gap in understanding the relationship between natural pollens and bee health outcomes.

RESPONSE: See RESPONSE above.

Generally I wouldn’t feel so strongly about a point not related to the results/ methods, but it really does border on misleading, and is not an appropriate interpretation of the data presented. You’ll see I reference this point a lot in the notes below – I don’t mean to beat a dead horse (though I really do hate this angle), but it just kept coming up – the theme of local / adaptive/ geographic was so important in the writing, and I am just trying to point out how prevalent this ‘local’/ ‘geographic’ language is used. At best, it could be a few lines in the discussion. If that is changed, and the other minor / clarifying changes are made, I think that this is a good paper!

RESPONSE: the theme of local / adaptive/ geographic was removed throughout the manuscript when referring to our study

Main results (from simple summary)

 Bees did not consume or show physiological responses that would indicate local adaptation to pollen

o This is technically a true statement, but the study design was not set up to address this question.

RESPONSE: The Simple Summary was rewritten to remove any statements referring to local adaptations.

 There were differences in responses, however, between workers from queens sourced from different geographic regions regardless of the pollen source they consumed.

o It is not shown that the region has anything to do with the variation – you are showing that bees from two suppliers were different – indicating that further study is needed to see either geographic regions / variability in suppliers. The study is also not designed to answer this question.

RESPONSE: We agree with this point, and changed the sentence to: ‘We detected differences between the queen lines in the amount of pollen they consumed and the size of their hypopharyngeal glands.’ (L28-29)

 There were also difference in responses depending on whether bees consumed pollen collected in the spring or the fall.

o This is also not in the study design, as the pollen was not from the same area in each collection

RESPONSE: The pollens were collected from the same areas of Arizona and Iowa in the spring and fall. They were not however, collected from the same areas where the queens were reared and open mated. The sentence was changed to:There also were seasonal differences between the nutrient composition of pollens. Spring pollens collected from colonies in both regions had higher amino and fatty acid concentrations than fall pollens.’ L29-31

While thy found seasonal differences in physiological responses to seasonal pollens, the differences did not align with the region where the bees and pollen were sourced.

o The study was also not set up to examine this as there is no indication of how the bees were selected and the pollen was not from the same region as the bees.

RESPONSE: I agree with the Reviewer and the statement ‘did not align with the region where the bees and pollen were sourced.’ was removed. (L-25-26)

Hypothesis: Nutrients available in seasonal pollens differ based on geographic region, and align with the seasonal activities of colonies

Methods:

Collected and analyzed pollen in the spring and fall from regions where brood rearing either stops in the fall (Iowa) or continues through the winter (Arizona).

- Fed both types of pollen to worker offspring of queens that emerged and open mated in each type of environment.

- Measured physiological response

o Pollen and protein consumption

o Gene expression levels (hex 70, hex 110, and vg)

o Hpyopharyngeal gland development

- Found differences in macronutrient content and amino and fatty acids between spring and fall pollens from the same region and differences in nutrient content between regions during the same season.

- Bees did not consume more pollen from their region of origin or provide evidence of being locally adapted in their responses (increased physiological metrics) to consuming local pollen.

- Queen type and seasonal effects in HPG size and difference in gene expression between bees consuming spring vs. fall pollen with larger HPG and higher gene expression levels in those consuming spring pollen.

- The effects might have emerged from the seasonal differences in nutritional content of the pollens and genetic factors associated with queen lines used.

General comment

1) I think that the wording of the hypothesis and some of the interpretation of the results is too broad/ general to the point of being confusing or deceptive (not maliciously so, just giving the wrong impression). When I read the summary and abstract I really thought it was going to answer some geographic questions, but the experimental design does not have the scope to do that. I would expect that a study that was really designed to answer this question would evaluate whether the inter-regional variation was greater than intra-regional variation, which this does not.

RESPONSE: I agree with the reviewer that the experimental design does not support comparisons on a regional basis. However, the study does support comparisons of nutritional composition of seasonal pollens collected by colonies and responses to workers from two different queen lines that were fed those pollens. The Simple Summary, Abstract and Discussion were extensively revised to narrow the findings from comparisons of pollen collected from two locations during spring and fall and fed to bees from two queen lines.

Examples where this language is used, and could be adjusted (I stopped, but there are many more points):

o Line 19 – we examined whether honey bees show local adaptation

RESPONSE: This line was deleted

o Line 32 – hypothesis that nutrients available in seasonal pollens differ based on geographic region

o Line 454 – beginning of the discussion –

Nutrients available in seasonal pollens differ based on geographic region.

 You do not answer this, as it would be incredibly possible/ possible to have an all brassica collection in IA in the spring.

RESPONSE: ‘geographic region’ is too broad a term for the pollen collected and fed to bees in this study. ‘Geographic region’ was changed to ‘location’ (e.g., ‘We examined responses of honey bees from two different queen lines fed pollens from locations that differed in floral species composition and yearly colony cycles.(L26-28), ‘We found differences in macronutrient content and amino and fatty acids between spring and fall pollens from the same location and differences in nutrient content between locations during the same season.’ (L-50-53)

This language would make me expect that you examined multiple regions in an area that had wintering and multiple regions in an area that didn’t have wintering. There is huge variation in blooms / available food within tiny geographic areas, and we would expect to see high variation within each state.

It is even more confusing because much of the collected pollen is not native to the region in which it is collected. That isn’t to say that it hasn’t been there since the bees have been there ( a few hundred years of pressure), but it does indicate that the pollen was collected in disturbed / replanted environment.

2) It needs to be really clear that this study is occurring in the United States, and not in Europe.

RESPONSE: USA was added to the locations where the pollens were collected and the studies were conducted (L-173,174,176, 177).

Many of the citations that are supporting the assertion for both local adaptation and for there to be a relationship between plants and the insects are from a place where these bees adapted. It is not necessarily true that you can directly extrapolate to the United States from these studies. I was really confused when reading this study why you would expect plants to be adjusted to the cycle of insects that have not been in the region for very long. If you do move forward with this angle, then this needs to be better supported why this is expected outside of the native range of this animal.

Examples:

o The paragraph starting on page 83 is not well supported. While the authors previous work does and excellent job showing that different pollens lead to different physiological changes, the link to geographical area is not strong. I think that the wording must be clear when you are describing areas where honey bees adapted (e.g. the Mediterranean, where citation 18 is from), and the United States, where it would not be expected or anticipated that plants and honey bees would have a strong connection since both occurred relatively recently, and have been highly dynamic since honey bees arrived.

RESPONSE: I agree with the reviewer that the argument for physiological changes in bees that are linked to geographic area is not supported in the broadest (i.e., regional) sense by our study. However, colonies that survive overwinter in confinement might do so, at least in part, because of the nutrients in the fall pollens they collect. If those nutrients were not available, colonies would not survive the winter and would not be established in those locations. Fall pollens in areas where colonies are active throughout the year might have different nutritional profiles than those where colonies are confined. The sentences in the paragraph were rewritten to clarify this point (‘Nutrients in fall pollens however, might be more variable because overwintering strategies differ depending on colony location.  For example, fall pollens in temperate regions might have high levels of lipids that can be stored in fat bodies of bees during winter confinement.’ (L 101-105). In our study, we did find that fall pollens from Iowa (where colonies overwinter in confinement) had higher lipid concentrations and significantly higher fatty acid levels than Arizona fall pollens. While we cannot say this trend exists in all areas where colonies overwinter in confinement, this was the case in our study. As stated in the Discussion, (L567-569) additional studies are needed to determine if the trend we detected in nutritional composition of seasonal pollens exists in other areas where colonies are confined during the winter.

o Evidence that local stock overwinters better : This would be clearer if the phrase “in Europe” was added. People use the term local stock in the US, but it technically not the same as local stock in Europe – where the bees have a long adaptive history and greater historical geographical separation.

RESPONSE: This information was added. The paragraph was rewritten for clarification (L119-133).

3) There is no support for the assertion that the bees from Iowa are adapted to a winter climate. (Don’t the Fassbinders do a lot of their queen rearing in Mississippi?). This is especially true for open mated queens. There is no way to say that the bees from California are any more or less adapted to shut down for winter than the bees that happened to be purchased from Iowa.

  1. If you wanted to make this assertion, then you would have to have survival / health evidence of these bees in the opposite locations.

RESPONSE: As stated earlier, we removed all references to our study testing for local adaptations. While Fassbinders might rear queens in Mississippi (I was not aware of this), we worked closely with Mr. Fassbinder to be sure that the queens we used in our study were from stock that overwintered in Iowa and were open mated in Fassbinder yards in Iowa. The reviewer makes a good point in that we cannot say if queens from California are more or less adapted to overwintering in Iowa than queens from Iowa. What can be said is that we compared responses in workers from queens reared and open mated in two in locations. Colonies established in the locations differ in overwintering strategies. The following changes were made to the paragraph; ‘If metabolism is a target for local adaptations that enhance fitness, then nutrients available in seasonal pollens and the bees’ responses to them might differ based on the location where the pollen is collected and queens are reared and mated. To test this, we collected pollen in spring and fall from colonies located in areas where colony overwintering strategies differ (Iowa and Arizona). The pollen was fed to worker offspring from queens reared and open mated in areas where bees are either confined overwinter (Iowa) or can rear brood throughout the year (California).’  (L134-141)

  1. There is HUGE genetic and behavioral variation in bees in northern states, including just within the state of Iowa. You don’t have to change the study here, but I would remove language that indicates that the Iowa bees are more likely to shut down for winter than the CA bees, since we don’t know that is true. It is still really valuable to test bees from two locations, but that is all that you can really say. I think the whole point about local adaptation is really a discussion point.

RESPONSE: The reviewer makes a good point; if California bees are hived in Iowa, they will go through an overwintering that differs from that in California. We removed all references to local adaptation and replaced them with comparisons between workers from two different queen lines reared and open mated in two locations and fed seasonal pollens from two locations. (L151-168)

Major comments:

Section 2.1: Please put the number of queens used in each group in this paragraph.

RESPONSE: The following information was added to the Materials and Methods: ‘Fifteen queens from each source were marked with a paint dot on their thorax prior to introducing them to their colony.’ L180-181. ‘Frames of sealed brood from all colonies headed by either IAq or CAq were placed in separate hive boxes according to queen source, and put in a temperature-controlled dark environmental room (32–34 °C, 30–40% relative humidity) [35].’ (L253-256)

Line 199: Please indicate how the frames of brood were chosen within the IA1 or CAq groups – were queens randomly chosen/ convenience? From how many hives were the frames taken? I think it would be useful to expand on the within state -genetic variation we are dealing with.

Response: We wanted to maximize genetic variation so we established 15 colonies each of CAq and IAq. We then took a sealed brood frame from each colony to emerge in an Environmental Room to supply bees for the cages. The following information was added to the Materials and Methods: ‘Frames of sealed brood from all colonies headed by either IAq or CAq were placed in separate hive boxes according to queen source, and put in a temperature-controlled dark environmental room (32–34 °C, 30–40% relative humidity) [35].’ (L253-256)

Line 206: Using the term “local pollen” for AZ pollen fed to CA bees is inappropriate. There is not enough support presented here or in the literature that the pollen in AZ is more ‘local’ than the pollen from IA. Stonyford, CA is about 1000 miles from Tucson. This is weird, but it also is really confusing to read. Strong recommendation to just call it IA pollen and AZ pollen. I’m super sympathetic to field research, and I’m sure there is a good logistical reason why CA pollen wasn’t used, but calling AZ pollen local for bees 1000 miles away, but not local for bees raised 1600 miles away is just weird.

- Recommended changes:

o Line 206

RESPONSE: All references to local and non-local pollens were removed from the manuscript.

o Line 388 – Heading should be reworded. Do bees consume more IA clover pollen compared to AZ brassica pollen? Or something general but true.

Response: The subheading was changed to, ‘Do workers respond differently to pollen mixes that differ in composition?  (L452-453)

‘ L447

Remove the term ‘local’ from this paragraph.

RESPONSE: I could not find the word ‘local’ in the paragraph that included L447 in the original manuscript (as opposed to the revised manuscript). That being said, the words local and non-local in reference to pollen were removed from the revised manuscript.

- Line 396 - This is super interesting, and I think is lost because you are stuck on the local story. I am so much more interested in just the biology of bees eating different pollens. If you just talked about brassica vs clover this paper would have soooo much more national significance since those are pollens that are super widespread all over the country (and Europe too, frankly). I would focus more on why you would see differential portioning of the pollen. They ate more AZ brassica pollen, but had higher hemolymph protein and higher HPG acini in IA clover pollen.

RESPONSE: The following information was included in the Discussion to address these points: In our study, lipid percentages and total fatty acid concentrations were within previously published ranges for bee collected pollen mixtures [54,55]. Palmitic, omega-3 and omega-6 were the predominant fatty acids found in AZp and IAp. These results are similar to previous reports of pollens collected by honey bees [19,54,56].  However, spring IAp (primarily Trifolium spp.) had significantly higher levels of omega-3 than AZp (Brassica spp.) and lower omega-6:3 ratios. Omega-3 is an essential fatty acid [57,58], and affects HPG development [52,59]. The higher omega-3 levels and lower omega-6:3 ratios in IAp may explain why CAq and IAq developed larger HPG when consuming IAp. This occurred despite consuming significantly less protein from IAp than AZp. Similar results were reported by [52], and suggest that HPG development might be more closely tied to lipid levels than to protein. This finding also indicates that Trifolium spp. could be an important pollen source for colony growth due to omega-3 levels and should be included in pollinator plantings. (L601-612)

- It is also really interesting that you see genetic differences in ug of protein consumed in fall. The way more interesting story line would be to try to find bees that utilize protein differently, and then bring those to the different areas as appropriate. Having differences in fall protein consumption makes me think only about how we could evaluate which version would be better for overwintering, and how we could screen for this type. (line 411)

RESPONSE: We agree that this was an interesting finding, and indicates that the two lines of bees we used are similar in that they consume more protein in the fall, but differ in how it might be used over the winter. In Arizona where the pollen was collected, some brood is reared throughout the year so some of the protein could be used for brood rearing. In Iowa where bees are confined, the protein is stored possibly in the bees’ HPG and perhaps used when the colony resumes brood rearing while still confined in the colony.

Line 291: The fact that the California local pollen is only brassica -which is not native in abundance (or maybe at all?) to either California or Arizona is a even more support for not using the term local. Tons of places, including Iowa have brassica in the spring.

RESPONSE: We agree, and removed local and non-local pollens from the manuscript.

Line 291: The fact that you are comparing mono-species pollen with mulitfloral pollen is a really big deal, and needs to be explicitly discussed.

RESPONSE: The following information was added to the Discussion to address the reviewer’s point: ‘Greater consumption of IAp might be due to the higher diversity of pollen types represented in IAp. Alternatively, higher consumption of IAp might be related to the ability of bees to digest the protein the pollen provided.’ (L618-621) and ‘Since spring pollens in our study were comprised primarily of Brassica spp. (AZp) and Trifolium spp., (IAp) our comparisons are in large part, between these two sources and between a monofloral (AZp) and polyfloral (IAp) mix. While spring AZp had higher protein concentrations than IAp, amino acid composition between the pollens was similar except for threonine which was higher in IAp. (L572-577)

Line 309: Spring AZp had significantly higher soluble protein that IAp == Is this Spring IAp or pooled / All IAp? Please clarify. Same for line 309 (fall). I’m pretty sure you mean spring and fall, but it is nice to leave no question.

RESPONSE: ‘spring’ was add L367.

This occurs again in line 355 – I would do a scan and just always have the qualifier for each pollen group.

RESPONSE: Additions of specific seasons also were made in L416 and 422.

Line 149: The methods here state the all the IA pollen was collected from Story county, but the caption for table 1 states that the Fall Iowa pollen was collected from Polk county.

RESPONSE: This correction was made in all figures. Iowa pollen was collected from Story County in the spring and Story and Polk counties in the fall (L361, 384, 403, 443)

Figure 2: IA pollen is labeled as “IO” pollen

RESPONSE IO was changed to ‘Iowa’ (L403). The Legend also was changed for consistency with Fig. 3.

Figure 4: SE bars missing from upper right subplot

RESPONSE: Great catch- error bars were added.

Table 1: While it can be easily calculated, it would be nice to report the values in ways that can be easily compared to bee health/ nutrition metrics. Specifically, including % protein and pollen to lipid ratios.

RESPONSE: The reviewer is correct that % protein can be easily calculated from the total soluble protein data in the table (e.g. 353.7 ug /mg = 35.3.7%) Adding this percentage would be showing the same data. The protein to lipid ratios were added to the Table and added to the Results section (L418-419).

Minor comments:

Line 28: The term “in temperate climates” (or some other indication that this is limited to Europe and North America) should be before the statement starting “In the spring…”, as this assertion is not true for many tropical climates, which are driven more by annual seasons of rain.

RESPONSE: This change was made (L38-39)

Lines 104-107: This statement requires a citation

RESPONSE: The citation was moved to the end of the paragraph (L133)

Figure 1: It would be nice if the clover was the same color for both the spring and the fall IA pollen (not necessary).

RESPONSE: The color for clover was changed for consistency. Note that the corrected version of Fig. 1 is in a separate file.

Line 308 – 311. I think that it would be easier to say that the pollen protein concentration was the same, except that Spring IA pollen was lower.

Spring AZ > Spring IAp

Fall AZ = Fall IAp

Spring AZ = Fall AZ

RESPONSE: The section was reworded; ‘Spring AZp had significantly higher soluble protein than spring IAp (Table 1). Protein concentrations were similar between fall AZp and IAp .’ (L367-368)

Similar for lines 355: Everyone is the same, but fall AZ p is lower.

Spring IAp = Spring AZp

Fall IAp > Fall AZ p

Spring IAp = Fall IAp

SpringAZ>Fall AZp

RESPONSE: This section also was reworded, ‘Pollen also was analyzed for lipid and fatty acid composition. Lipid concentrations were similar between spring IAp and spring AZp, but higher in fall IAp than fall AZp (Table 1).  Spring and fall IAp had similar lipid concentrations.  AZp had higher lipid concentrations in the spring than the fall.’ (L415-419)

Line 668: The citation has the title listed twice. 6

RESPONSE: The citation was corrected (L789-790)

Discussion:

- All of the pollen is high in protein – there may be no differences because it is above a threshold.

RESPONSE: I’m not entirely sure what the reviewer is referring to here. We did detect difference in soluble protein levels specifically between spring AZp and IAp. The pollen samples were sufficiently diluted so that the protein levels in the pollen were not above the threshold for our assay used to measure protein concentration.

- Single source pollen compared to multifloral pollen (barely).

o Almost a study of brassica vs clover vs Woolly Bur-sage.

RESPONSE: The point that the spring pollen could be seen a comparison between clover and Brassica was included in the Discussion (L572-575). I hesitate to make a similar comparison between the fall pollens that had other species in higher frequency than in the spring pollens.

Round 2

Reviewer 1 Report

I have reviewed the revised version of the MS. I believe that it is ready to roll with few minor tweaks. 

___

1. races/subspecies: I believe »subspecies« replaced the word »race« (which is supposed to be out of favor for some time now).

2. unclear comment from previous review:

Sup tables: ensure that the individual table fits on single page with all the comments below

RESPONSE: I am not sure what the reviewer is referring to here.

RESPONSE TO RESPONSE: in my file, the tables spread over to more than one page, making it difficult to read. I am sure that with decreasing the fonts a little it would be possible to squeeze every table to one page and still keep it readable. Tables S2-S5.

Author Response

'race' was changed to 'subspecies'. Each of the supplementary tables (S2,S3,S4 and S5) were edited to fit on a single page.

Reviewer 2 Report

The authors sufficiently addressed all of the comments. 

Author Response

No suggested changes.